# New insights into the plasma and urinary metabolomic signatures of spontaneously hypertensive rats

Celia Rodríguez-Pérez[1,2,3☉], Alejandra Vázquez-Aguilar[2,4☉],
Oscar Daniel Rangel Huerta[5,6*], Estefanía Sánchez-Rodríguez[2,3,4,7], Ángel Gil[2,3,4,7],
Caridad Díaz[8], Félix Vargas[9], María D. Mesa[2,3,4,10]

1 Department of Nutrition and Food Science, Faculty of Pharmacy, University of Granada, Granada, Spain, 2 Institute of Nutrition and Food Technology (INYTA) 'José Mataix', Biomedical Research Centre, University of Granada, Granada, Spain, 3 Instituto de Investigación Biosanitaria ibs. GRANADA, Granada, Spain, 4 Department of Biochemistry and Molecular Biology II, Faculty of Pharmacy, University of Granada, Granada, Spain, 5 Section of Chemistry and Toxinology, Norwegian Veterinary Institute, Ås, Norway, 6 SpectraMinds, Nordre Follo, Norway, 7 CIBEROBN (CIBER Physiopathology of Obesity and Nutrition CB12/03/30028), Institute of Health Carlos III (ISCIII), Madrid, Spain, 8 Department of Screening & Target Validation, Fundación MEDINA, Granada, Spain, 9 Department of Physiology, Faculty of Medicine, University of Granada, Parque Tecnológico de la Salud, Avenida de la Investigación, Armilla, Granada, Spain, 10 Spanish Network in Maternal, Neonatal, Child and Developmental Health Research (RICORS-SAMID), RD24/0013/0007, Instituto de Salud Carlos III (ISCIII), Madrid, Spain

☉ Joint first authors
* odrangel@spectraminds.io

## Abstract

### Background

Hypertension is a major risk factor associated with cardiovascular diseases and one of the leading causes of premature death. Metabolomics is a useful tool for studying in vivo metabolic profiles to better understand the pathogenesis of diseases such as hypertension. This work aimed to explore the plasma and urinary non-targeted metabolic profile of 16-week-old spontaneously hypertensive rats (SHR) to identify new metabolomic profiles associated with hypertensive phenotypical characteristics.

### Methods

Plasma and 24-hours urine samples were collected from 10 SHR and 10 age-matched normotensive Wistar-Kyoto 16-week-old male rats. Plasma and urinary metabolic profiles were investigated using high-performance liquid chromatography quadrupole time of flight coupled to mass spectroscopy followed by multivariate statistical analysis. The *mummichog* pathway enrichment analysis was used to integrate metabolomics data into biological contexts.

**Data availability statement:** All processed data from the metabolomics datasets is available at https://doi.org/10.5281/zenodo.15628770.

**Funding:** This work was funded by Programa Operativo FEDER 2014-2020/ Junta de Andalucía-Consejería de Economía y Conocimiento/ Proyecto (B-AGR-257-UGR18); and by the Ministry of Economy, Industry and Competitiveness of Spain and the Junta and Andalucía, through the FEDER INNTERCONECTA Program of the Center for Industrial Technological Development (CDTI) -"CARDIOLIVE StudyProject No. ITC-20151142 (EXP 00083147), co-financed by the European Regional Development Fund (FEDER), and SAN FRANCISCO DE ASIS DE MONTEFRÍO S. Coop. RICORS funded by the Recovery, Transformation and Resilience Plan 2017-2020, ISCIII, and by the European Union – NextGeneration EU, ref. RD21/0012/0008, and RICORS funded by the Recovery, Transformation and Resilience Plan 2021-2024, ISCIII, and by the European Union – NextGeneration EU, ref. RD24/0013/0007. The funders had no role in study design, data collection and analysis, decision to publish, or preparation of the manuscript.

**Competing interests:** The authors have declared that no competing interests exist.

## Results

A total of 16 differential metabolites were found in plasma and 13 differential metabolites in urine from SHR compared to normotensive rats. Differences in some microbiota-derived metabolites suggest changes in the gut microbiota associated with hypertension in our experimental model. The *mummichog* algorithm has recognized that hypertensive metabolism is associated with the altered metabolism of steroid hormones, bile acid, and purines.

## Conclusions

This work highlights the importance of metabolomics as a tool for the identification of biomarkers related to hypertension and its consequences in SHR. The findings suggest that alterations in the metabolism of steroid hormones, bile acids, and purines, as well as metabolites derived from the intestinal microbiota are associated with the presence of hypertension. More research is needed to further understand their role during hypertension.

## Introduction

Arterial hypertension is a complex multifactorial vascular pathology caused by the interactions between environmental and polygenic factors. It is associated with other adverse cardiovascular complications, including myocardial infarction, stroke, kidney disease, and global mortality, and has been reported as one of the leading causes of premature death, and the world's most prevalent cardiovascular disorder, affecting 1.28 billion adults aged 30–79 years worldwide [1]. However, despite the availability of several preventive and therapeutic approaches, arterial hypertension continues to be an unresolved risk factor for disease burden worldwide [2]. Therefore, understanding new mechanisms related to the pathogenesis of hypertension and the onset of its complications may help identify alternative preventive and therapeutic tools.

Among different preclinical hypertensive animal models, spontaneously hypertensive rats (SHR) are the most widely used genetic models of hypertension for the study of primary or essential hypertension owing to their similarities to human hypertension [3,4]. The systolic blood pressure (SBP) of SHR reaches 180–200 mm Hg after 4 weeks of growth, while their breeding brother Wistar-Kyoto rats (WKY) remain normotensive. The SHR model has been used to identify hypertension-related genes, evaluate targeted organ complications, and screen of potential pharmacological drugs; indeed, they are expected to provide insight into pathological and therapeutic mechanisms for the regulation of blood pressure. Interestingly, hypertension has also been linked to immune disorders and dysbiosis [5], but its molecular mechanisms have not been fully elucidated and require further investigation [6].

Metabolomics is an emerging discipline that characterizes the small molecules (< 2000 Da) present in a biological sample that are closely related to an organism's phenotype, providing wide information about biological systems and metabolism.

However, it requires the use of robust bioinformatics techniques and statistical strategies to mine large amounts of data to extract relevant biological information, as well as other novel tools, such as *mummichog,* oriented to predict metabolic patterns that may be associated with a disease, rather than the individual identification of metabolites [7].

The present study aimed to compare *in vivo* plasma and urine metabolic differences between hypertensive 16-wks age SHR and normotensive healthy WKY rats, using liquid chromatography-mass spectrometry non-targeted metabolomic strategies, to identify possible metabolic pathways that may be altered in the presence of hypertension, and could be related to hypertension development or to its pathological consequences.

## Materials and methods

### Animals and study design

A total of 20 (n = 10 SHR and n = 10 Wistar-Kyoto (WKY)). Animals were purchased at 8 weeks of age from Janvier Labs (CEDEX, France). All the rats had *ad libitum* access to food and water. The animals were fed a standard maintenance diet (Panlab) containing barley, wheat, maize, soybean meal, wheat bran, hydrolyzed fish proteins, dicalcium phosphate, a pre-mixture of minerals, calcium carbonate, a pre-mixture of vitamins, 68.9% carbohydrates (fiber 3.9%), 16.1% proteins, and 3.1% fat, during 8 weeks of follow up, and were euthanized at the age of 16 weeks. No differences in food or water intake were observed between the two groups. Animal experiment was performed in accordance with the guidelines set by the European Community Council Directives for the Ethical Care of Animals (86/609/EEC) and approved by the Ethics Committee of Laboratory Animals of the University of Granada (Spain, permit number 18/07/2017/099). This study was conducted according to the ARRIVE guidelines.

SBP was monitored to evaluate the state and evolution of the disease. SBP was measured by plethysmography in conscious rats (LE 5001-Pressure Meter, Letica SA, Barcelona, Spain). Animals were placed in individual restrainers that allow immobilization without the use of anesthesia and after habituation to this practice during several days prior to the measurement. At least seven determinations were made at every session, and the mean of the lowest three values within a range of 5 mmHg was set as the final SBP value. The rats were introduced into metabolic cages (Panlab, Barcelon, Spain) for 24-h urine collection. Food and water intake were monitored daily by weighing the remaining food or by measuring the remaining volume of water, respectively, allowing the calculation of daily food and water consumption. The present study was conducted in accordance with the ARRIVE (Animal Research: Reporting of In Vivo Experiments) guidelines. All efforts were made to minimize animal suffering through constant clinical monitoring and environmental enrichment. Chemical restraint was achieved via intraperitoneal (i.p.) administration of Equitensin (2.5 mL/kg). Once a surgical plane of anesthesia was confirmed, blood samples for plasma variable determination were collected via abdominal aortic puncture using a beveled needle for fractional extraction. Finally, rats were euthanized by exsanguination while under deep anesthesia. Blood was centrifuged at 1750 × g for 10 min at 4°C. Aliquots of plasma were frozen immediately at −80°C until metabolomic analysis. In addition, after euthanasia, necropsy was performed and the heart, kidneys, liver, and tibia were collected and their absolute weights recorded to evaluate animal development and organ-specific involvement.

### Metabolite extraction

Plasma and urine samples were thawed on ice and kept at 4°C during the entire extraction process and pretreated as described by Liu et al. with some modifications [8]. Briefly, 100 μl of plasma samples were deproteinized with 200 μl of a precipitation mixture containing acetonitrile, methanol, and acetone (8:1:1 v/v), vortexed, and placed at −20°C for 30 min to facilitate protein precipitation. The samples were centrifuged at 14800 rpm for 10 min at 4°C, and the supernatants were evaporated in an Eppendorf® Concentrator Plus evaporator centrifuge for 2 h. Dried samples were reconstituted in 100 μl of 0.1% formic acid in water, vortexed for 20 s, and incubated in an ice bath for 10 min. Subsequently, the samples were centrifuged at 14800 rpm for 10 min at 4°C, and 40 μl of the supernatant was transferred to a high-performance liquid

Chromatography (HPLC) vial with 250 µl inserts. Five microliters of internal standard (IS) consisting of a mixture of roxi-thromycin 3 ppm, tryptophan-D5 5 ppm, hydroxydiclofenac 5 ppm, warfarin 3 ppm, ibuprofen 5 ppm, and bisphenol 3 ppm, were added prior the extraction [9]. The IS allowed the monitoring of instrument performance and aided chromatographic alignment. The prepared samples were stored at −80°C until their analysis.

In the case of urine, after measuring osmolarity (Gonotec Osmomat 030 D Osmometer) and adjusting for differences in hydration status among animals, all samples were diluted to 200 ± 20 mOsm/kg using Milli-Q water [10]. The samples were then centrifuged at 14800 rpm for 10 min at 4°C, 40 µl of the supernatant was transferred to an HPLC vial, and 5 µL of the abovementioned IS was added. The prepared samples were stored at −80 °C until analysis.

Quality control (QC) samples from plasma and urine were prepared by pooling equal volumes (20 µl) of all studied samples. Blanks were prepared using water.

### HPLC-ESI-QTOF-MS analysis

Samples were analyzed using an Agilent Series 1290 High Performance Chromatography (Agilent Technologies, Santa Clara, CA, USA) coupled to an AB SCIEX TripleTOF 5600 quadrupole time-of-flight mass spectrometer (qTOF-MS), using a Waters Atlantis T3 C18 chromatographic column (2.1 mm x 150 mm, 3 µm, Waters Corporation, Milford, MA, USA), maintained at 30°C in the oven. The mobile phases used were 0.1% formic acid in a mixture of water (90:10) (A) and 0.1% formic acid in MeCN: water (90:10) (B). The column was eluted with the following gradient: 0 to 0.5 min, 0% eluent B, 0.5 to 11 min, 100% eluent B, which was maintained until 15.60 min and 0% of eluent B and 15.60–20 min 0% eluent B at a constant flow rate of 0.3 mL/min. The injection volume was 5 µl.

Triple TOF 5600 operating in positive- and negative-ion modes was employed for metabolite detection using a mass range of 50–1250 Da. The Triple TOF used a Duo Spray source with separate electrospray ionization (ESI) and atmospheric-pressure chemical ionization probes. The ESI parameters were as follows: positive mode, capillary voltage of 5000 V, nebulizer gas pressure of 50 psi, drying gas pressure of 50 psi, temperature of 500°C, and focusing potential of 100 V. Negative mode: capillary voltage −4500 V, nebulizer gas pressure, 50 psi; drying gas pressure, 50 psi; tempera-ture, 500 °C; focusing potential, 100 V. The eight most intense ions in each cycle were fragmented.

To avoid possible bias the sample injection sequence was randomized, and QC samples were injected at the start of the analysis and every five samples to stabilize the instrument conditions, monitor the system performance and adjust the signal intensity drift within and between the batches.

### Data acquisition and analysis

The MarkerView software (version 1.2.1, AB SCIEX, Concord, ON) was employed for data set creation with the following extraction parameters: retention time (RT) 1.20–15.00 min; offset subtraction 10 scans; Subtraction Mult. Factor 1.3, noise floor 50 counts per second (cps), minimum spectral peak width 0.02 Da; minimum peak width RT (4 scans); RT tolerance (0.12 min); mass tolerance (10 ppm); required number of samples (6), and maximum number of peaks at 5000. After align-ing the data, anomalous results were filtered, and a raw data matrix was obtained.

### Data preprocessing

From the raw data matrix, another matrix was extracted with RT, the mass-charge ratio (m/z), and the areas of metabolic characteristics detected. Alternative non-parametric measures of the relative standard deviation and the D-ratio were calculated with a cut-off point of 0.2 and 0.4, respectively. For missing values, imputation was performed using the ran-dom forest algorithm. All calculations were performed using the Notame package in R (v0.0.900) [11]. Subsequently, the data were evaluated to identify technical drift, which was corrected by modeling using a spline-cubic regression based on the QC samples and then corrected for the abundance of all samples by reversing the modeling effect of the drifts. This process was developed independently for each feature.

## Multivariate and univariate statistical analyses

Multivariate statistical analyses were performed using SIMCA-P (V16; Sartorius Stedim Biotech, Umeå, Sweden) following the workflow described by Rangel-Huerta et al. [12]. Datasets were scaled by Pareto scaling. Principal Component Analysis (PCA) was employed to assess the quality of the metabolomic data on a QC basis to discard potential outliers, identify clustering patterns, and visualize the total variation of the metabolite profiles. Orthogonal Partial Least Squares Discriminant Analysis (OPLS-DA) was used to identify metabolite patterns and specific features that could discriminate between groups. Seven-round cross-validation and 100 permutations were applied to this model, and subsequently, cross-validation analysis of variance (CV-ANOVA) was performed to assess the reliability of the models. Statistical significance was tested at a p-value $\leq 0.05$. Additionally, R2X and Q2 were evaluated to assess robustness, considering that values close to one reflect a reliable model. Variable selection of the most relevant features was performed using an S-plot, considering $p > 0.05$, $p(corr) > 0.5$, and an adjusted p-value from a t-test ($p < 0.05$) was considered significant.

## Pathway analysis and metabolite identification

The MS Peaks to Pathways module from MetaboAnalyst was employed to predict the pathway activity from untargeted mass spectral data was employed [13]. To this end, *p*-values and *t*-scores were determined for all metabolites in both the plasma and urine samples of the SHR and WKY groups. The analysis was performed in mixed mode (positive and negative) with the following parameters: 1) instrument mass precision of 10 ppm, 2) retention time (RT) in minutes, 3) p-value, 4) positive/negative analytical mode, and 5) *Mummichog* algorithm with a cut-off value of $p \leq 0.05$ and gene set enrichment analysis (GSEA). To minimize false positives, pathways with an overlap size smaller than three were excluded from the analysis. *R. norvegicus* was selected from the KEGG pathway map for analysis. Pathways with a combined p-value (*Mummichog* and GSEA) $\leq 0.1$ were considered significant.

Subsequently, KEGG GENOME (<https://www.genome.jp/kegg/kegg2.html>) was accessed using the codes of the compounds previously identified in MetaboAnalyst by applying a filter of sets of metabolites associated with the route, and the chemical compounds were annotated. Molecular annotation of the relevant compounds obtained in the multivariate analysis was carried out in SIRIUS 6.3.3 [14] and public databases using MS-DIAL software [15] or mzmine 4.8.5 [16]. The identity of the metabolites of interest was confirmed by considering the mass accuracy and true isotopic pattern in both the MS and MS/MS spectra provided by qTOF-MS.

# Results

## Animal description

The SBP, heart weight, and left ventricular weight were higher in SHR *vs* WKY rats, as well as water intake and diuresis, and heart rate tended to be higher ($p = 0.054$) (Table 1). The 24-h food intake was similar in SHR, as well as tibia length and kidney weight, while the body and liver weights were lower in SHR than in WKY rats (Table 1).

## LC-qTOF-MS untargeted metabolomics analysis

After pre-processing, quality assessment, data filtering, and clustering, a dataset containing 195 features in the positive mode and 174 in the negative mode for plasma and 162 features in the positive mode and 239 in the negative mode for urine was used for further statistical analysis.

An unsupervised multivariate statistical approach was used to compare the metabolomic differences between the SHR and WKY rats. The PCA score plots of plasma and urine (Fig 1A and 1B, respectively) indicated a good separation between the SHR and WKY animal models. The score plot for the first two principal components did not reveal any outliers.

**Table 1. Food and water intakes, body weight, diuresis, SBP and heart rate, and the tibia length, kidney weight, liver weight, heart weight, and the left ventricular weight, of the 16 weeks-old SHR and control Wistar-Kyoto rats.**

| | SHR | | | Wistar | | | |
|---|---|---|---|---|---|---|---|
| | **Mean** | **±** | **SEM** | **Mean** | **±** | **SEM** | ***P* value** |
| Food intake (g/d) | 18.8 | ± | 1.2 | 18.1 | ± | 1.0 | 0.333 |
| Body weight (g) | 403.4 | ± | 9.7 | 444.9 | ± | 4.0 | <0.010 |
| Water intake (ml/d) | 29.4 | ± | 2.6 | 21.4 | ± | 2.8 | 0.025 |
| Diuresis (ml/d) | 11.1 | ± | 0.8 | 6.1 | ± | 0.3 | <0.001 |
| SBP (mmHg) | 197 | ± | 4 | 146 | ± | 3 | <0.001 |
| Heart rate (bpm) | 349 | ± | 4 | 337 | ± | 6 | 0.054 |
| Tibia length (mm) | 49.9 | ± | 0.2 | 49.8 | ± | 0.3 | 0,404 |
| Kidney weight (g) | 1.25 | ± | 0.04 | 1.30 | ± | 0.02 | 0.157 |
| Liver weight (g) | 13.7 | ± | 0.6 | 15.0 | ± | 0.3 | 0.034 |
| Heart weight (g) | 1.33 | ± | 0.03 | 1.05 | ± | 0.01 | <0.001 |
| Left ventricular weight (g) | 1.12 | ± | 0.02 | 0.86 | ± | 0.01 | <0.001 |

Data are presented as mean ± SEM. P value indicates differences between groups. Student T-test was used to compare normal distribution variables. Bpm, beats per minute; SBP, systolic blood pressure; SEM, standard error of the mean; SHR, spontaneously hypertensive rats; WKY, Wistar Kyoto healthy rats.

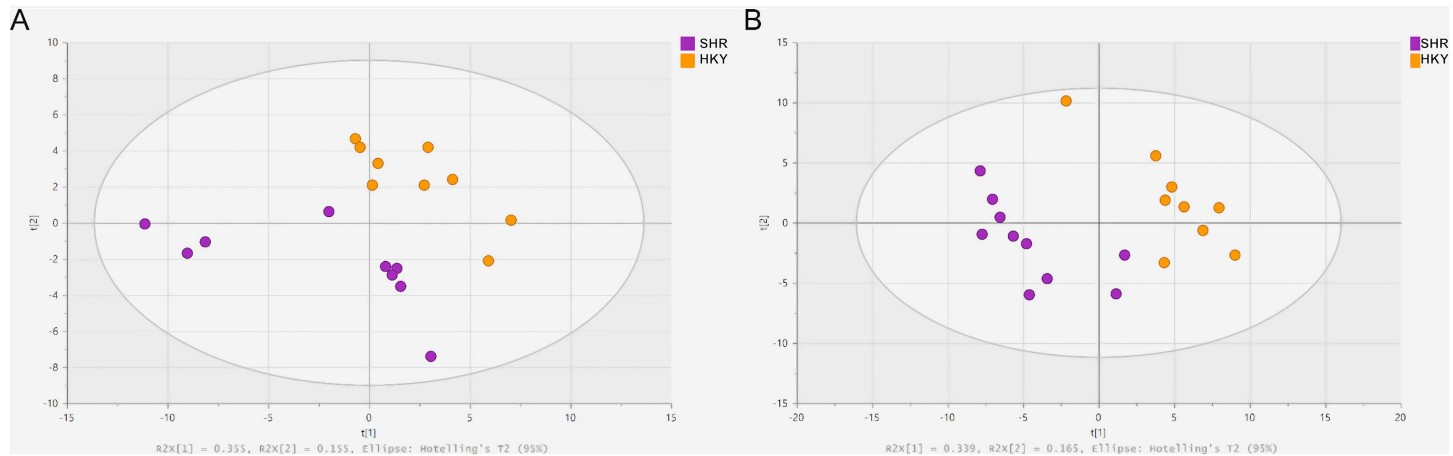

**Fig 1. PCA scores plot from LC-qTOF-MS data for the plasma (A) and urine (B) of SHR and WKY animal models.** Separation along the Y-axis represents variation between animal models. LC-qTOF-MS, Liquid Chromatography coupled to quadrupole time-of-flight mass spectrometer. PCA, principal component analysis; SHR, spontaneously hypertensive rats; WKY, Wistar Kyoto healthy rats.

Supervised multivariate analysis with OPLS-DA indicated a strong separation between the two groups (Scores plots are included as S1 File). One hundred permutation tests were conducted to evaluate the statistical robustness of plasma and urine OPLS-DA models. Good discrimination was observed between the two groups with values obtained for the plasma model: $R^2X$ (0.504), $R^2Ycum$ (0.954), $Q^2cum$ (0.874), and CV-ANOVA of 9.38E-06, and for the urine model: $R^2X$ (0.572), $R^2Ycum$ (0.977), $Q^2cum$ (0.837), and CV-ANOVA of 0.00253, thus indicating that the models were reliable and provided excellent prediction ability (Permutations plots are included as in S1 File). Additionally, the S-plot derived from the OPLS-DA models depicts a scatter plot that combines the modeled covariance (X-axis) and modeled correlation

(Y-axis) from OPLS-DA, allowing the identification of interesting variables. Fig 2A and 2B show plasma and urine S-plots, respectively. Variables showing p[1] > 0.05 and p(corr) > 0.5 values were considered the most relevant metabolites for the differentiation between samples.

Specific observations were made on the differential metabolites identified in the SHR compared to those in WKY rats. Of the 59 and 111 differential metabolites in plasma and urine, respectively, 16 and 13 were annotated in plasma and urine, respectively. These metabolites are presented in Table 2 and Table 3, respectively, indicating their RT, observed m/z, suggested ion, fragments from MS/MS experiments, probability, p(corr) value from S-plots, annotation, and family class. Spectral matching plots are included in S1 File.

In plasma, two p-cresol derivatives, two adducts of cholic acid, and twelve glycerophospholipids, including nine phosphatidylcholines and three phosphatidylethanolamines, were significantly different in SHR compared to WKY, while in urine, 13 compounds were identified: one nucleoside, (pseudouridine); one organoheterocyclic compound (uric acid), four organic acids and derivatives, four organic oxygen compounds, one benzenoid, and two phenylpropanoids and polyketides.

Finally, the results of the pathway enrichment analysis of plasma and urinary metabolites significantly associated with SHR are shown in Fig 3A and 3B. These analyses show significant or borderline significant enrichment of several pathways related to hypertension in plasma, namely steroid hormone biosynthesis (combined $p$ value = 0.0049), primary bile acid biosynthesis (combined $p$ value = 0.063), and arachidonic acid metabolism (combined $p$ value = 0.084) for plasma and purine metabolism in urine (combined $p$ value = 0.0071).

## Discussion

This study deepens the understanding of plasma and urine metabolic differences between 16-week-old hypertensive SHR and aged-matched normotensive WKR rats using LC-qTOF-MS non-targeted metabolomics approaches. Sixteen plasma and 13 urine metabolites were semiquantitatively found to be different after deconvolution and were tentatively annotated as a specific compound or as a chemical class. In addition, *mummichog* pathway enrichment

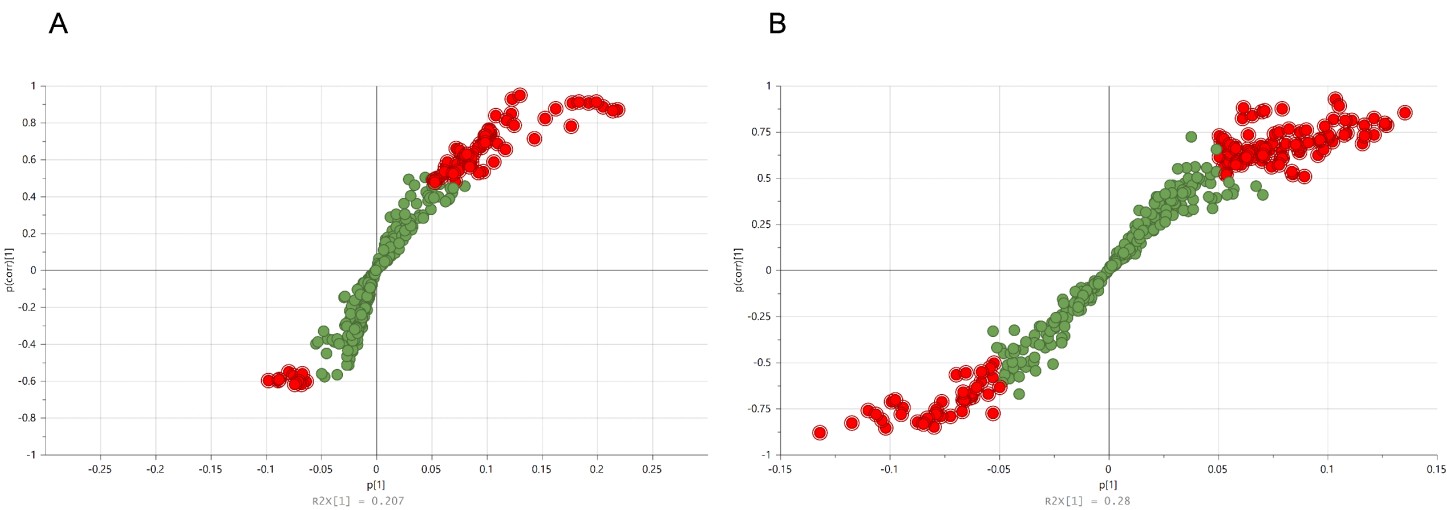

**Fig 2. S-plot derived from the OPLS-DA model indicating plasma (A) and urine (B) biomarkers increased in WKY (upper right) and SHR (lower left).** Only variables with p > 0.05 and p(corr) > 0.5 were considered significant and presented in the plot. OPLS-DA, orthogonal partial least squares discriminant analysis; SHR, spontaneously hypertensive rats; WKY, Wistar Kyoto healthy rats.

**Table 2. Plasma identified discriminant metabolites between SHR and WKY rats.**

| RT (min) | Observed m/z | Suggested ion | Fragments | Proba-bility* | p(corr) | Direction | Suggested annotation[1] | Family class |
|---|---|---|---|---|---|---|---|---|
| | | | **Negative ionization mode** | | | | | |
| 4.83 | 283.083 | [M – H]⁻ | 75.0077, 113.0226, 85.0293, 57.0337 | 100% | 0.67 | ↑ in SHR | p-Cresol glucuronide | Organic acids and derivatives |
| 6.54 | 187.007 | [M – H]⁻ | 107.0500 | 100% | 0.06 | ↑ in SHR | p-Cresol sulfate | Organic acids and derivatives |
| 8.75 | 407.280 | [M – H]⁻ | 389.2711, 345.2808 | 100% | 0.80 | ↑ in SHR | Cholic acid | Steroids and derivatives |
| 10.26 | 538.313 | [M + CH2O2 - H]⁻ | 478.2946, 253.2185 | 98% | 0.75 | ↑ in SHR | LPC (16:1) | Glycerophospholipids |
| 10.69 | 524.278 | [M – H]⁻ | 327.2322, 283.2428, 214.0489 | 93% | 0.61 | ↑ in SHR | LPE (22:6) | Glycerophospholipids |
| 11.04 | 552.331 | [M + CH2O2 - H]⁻ | 492.3098, 267.2335 | 99% | 0.70 | ↑ in SHR | LPC (17:1) | Glycerophospholipids |
| 11.32 | 590.349 | [M + CH2O2 - H]– | 530.3276, 305.2507 | 90% | 0.60 | ↑ in SHR | LPC (20:3) | Glycerophospholipids |
| | | | **Positive ionization mode** | | | | | |
| 8.77 | 426.321 | [M + H3N + H]⁺ | 355.2643, 373.2751 | 100% | 0.82 | ↑ in SHR | Cholic acid | Steroids and derivatives |
| 9.83 | 468.309 | [M + H]⁺ | 184.0743, 450.2996, 104.1072 | 100% | 0.64 | ↑ in SHR | LPC (14:0) | Glycerophospholipids |
| 10.04 | 494.324 | [M + H]⁺ | 184.0743 | 91% | 0.78 | ↑ in SHR | LPE (16:1) | Glycerophospholipids |
| 10.06 | 542.326 | [M + H]⁺ | 184.0736, 524.3126, 104.1066 | 96% | 0.60 | ↑ in SHR | LPC (20:5) | Glycerophospholipids |
| 10.26 | 494.323 | [M + H]⁺ | 476.3156, 184.0746, 104.1080 | 99% | 0.74 | ↑ in SHR | LPC(16:1) | Glycerophospholipids |
| 11.04 | 508.342 | [M + H]⁺ | 184.0729, 490.3279 | 98% | 0.72 | ↑ in SHR | LPC(17:1) | Glycerophospholipids |
| 11.53 | 546.354 | [M + H]⁺ | 184.0746, 528.3466, 104.1076 | 94% | 0.63 | ↑ in SHR | LPC (20:3) | Glycerophospholipids |
| 11.72 | 480.309 | [M + H]⁺ | 339.2902, 462.2994, | 100% | 0.67 | ↑ in SHR | LPE (18:1) | Glycerophospholipids |
| 12.03 | 572.372 | [M + H]⁺ | 184.0738, 554.3624, 104.1072 | 92% | 0.65 | ↑ in SHR | LPC (22:4) | Glycerophospholipids |

LPC, lysophosphatidylcholines; LPE, Lysophosphatidylethanolamine; m/z mass to charge ratio; PE, phosphatidylethanolamine; RT retention time. *Corresponds to class assignment. Only those compounds that had MS/MS information have been included.

[1]All the annotations included are Level 2 according to the Metabolomics Standards Initiative

analysis has been employed to integrate metabolomic data into biological contexts [17]. The *mummichog* algorithm recognizes specific metabolic pathways that act differentially in SHR and normotensive Wistar rats, mainly steroid hormones, from plasma and urine metabolites, and purine metabolism, from urine metabolites, in accordance with the individual compound annotation. In addition, gut microbiota and primary liver bile acid metabolism were altered in hypertensive rats.

The *mummichog* algorithm predicts the functional activities of metabolites and enables metabolic pathway-level analyses by searching for chemical identities, without the need to identify individual compounds. Therefore, we use this approach to contextualize the observed changes, which are initially 'clouded' by the presence of other, more abundant metabolites, such as LPCs, identified in the multivariate analysis. In sterol hormone biosynthesis pathway, *mummichog* pathway analysis identified several plasma metabolites, indicating that this metabolic pathway is affected in SHR. Specific types of hypertension caused by different forms of inherited mineralocorticoid pathway defects have been described [18,19]. In addition, a previous study reported that serum steroid hormone, progesterone, corticosterone, and cortisol concentrations differed significantly between SHR older than 10 weeks and their age-matched WKY rats, suggesting a role for progesterone in the development of hypertension [20], independent of other sex hormones [21]. This metabolic alteration may be implicated in stress-derived hypertension complications [22]. Therefore, preventive hypertension follow-up should be considered in patients with complications in steroid hormones biosynthesis to control the development of this disease and its complications.

**Table 3. Urine identified discriminant metabolites between SHR and WKY rats.**

| RT (min) | Observed m/z | Suggested ion | Fragments | Probability* | p(corr) | Direction | Suggested annotation[1] | Family class |
|---|---|---|---|---|---|---|---|---|
| | | | **Negative ionization mode** | | | | | |
| 1.43 | 243.062 | [M − H]⁻ | 183.0410, 124.0259 | 98% | −0.66 | ↑ in Control | Pseudouridine | Nucleoside and nucleotide analogues |
| 1.47 | 167.0211 | [M − H]⁻ | 124.0148 | 99% | −0.79 | ↑ in Control | Uric acid | Organoheterocyclic compounds |
| 1.65 | 191.0198 | [M − H]⁻ | 173.0090, 103.0397 | 94% | −0.52 | ↑ in Control | Citric acid | Organic acids and derivatives |
| 2.58 | 357.0839 | [M − H]⁻ | 339.0720, 181.0502, 175.0250, 137.0602, 113.0237 | 100% | 0.65 | ↑ in SHR | DHPPA (A-D-glucuronide) | Organic oxygen compounds |
| 3.55 | 326.0894 | [M − H]⁻ | 175.0250, 150.0560, 108.0447, 113.0235 | 100% | 0.50 | ↑ in SHR | Unknown | Organic oxygen compounds |
| 4.36 | 178.0518 | [M − H]⁻ | 160.0394, 148.0406, 134.0613, 77.0393 | 100% | 0.76 | ↑ in SHR | Hippuric acid | Benzenoids |
| 4.78 | 193.0513 | [M − H]⁻ | 178.0280, 149.0610, 134.0376 | 96% | 0.76 | ↑ in SHR | Ferulic acid | Phenylpropanoids and polyketides |
| 4.78 | 273.0091 | [M − H]⁻ | 193.0507, 178.0275, 149.0608, 134.0371 | 97% | 0.80 | ↑ in SHR | Ferulic acid-4-O-Sulfate | Phenylpropanoids and polyketides |
| 4.86 | 188.9870 | [M − H]⁻ | 109.0292 | 99% | −0.51 | ↑ in Control | Resorcinol monosulfate | Organic acids and derivatives |
| 4.89 | 283.0840 | [M − H]⁻ | 107.0498, 175.0253, 157.0141, 113.0244, 85.0288 | 100% | 0.70 | ↑ in SHR | p-Cresol glucuronide | Organic oxygen compounds |
| 5.18 | 231.0797 | [M − H]⁻ | 74.0243, 156.0457, 128.0501 | 100% | 0.67 | ↑ in SHR | Indole-3-acetyl-glycine | Organic acids and derivatives |
| 5.20 | 171.1036 | [M − H]⁻ | 127.1124, 11.0804 | 100% | −0.78 | ↑ in Control | 8-oxononanoate | Organic acids and derivatives |
| 5.70 | 297.0994 | [M − H]⁻ | 121.0653, 175.0254, 113.0242, 85.0287 | 96% | 0.56 | | 4-Ethylphenol glucuronide | Organic oxygen compounds |
| | | | **Positive ionization mode** | | | | | |
| 4.66 | 180.065 | [M + H]+ | 105.0327, 77.0377 | 100% | 0.70 | ↑ in SHR | Hippuric acid | Benzenoids |

DHPPA Dihydroxyphenylpropionic acid; m/z mass to charge ratio; RT retention time; TCA, tricarboxylic acids. *Corresponds to class assignment. Only those compounds that had MS/MS information have been included.

[1]All the annotations included are Level 2 according to the Metabolomics Standards Initiative

*Mummichog* pathway analysis revealed altered purine metabolism in SHR, including increased urinary levels of uric acid, allantoic acid, and (S)-allantoin. Uric acid is the final oxidation product of purine metabolism in humans; however, in most mammals, such as rats, the hepatic uricase oxidizes uric acid to allantoin, which is subsequently excreted in the urine. Epidemiological studies have demonstrated a significant association between hyperuricemia and hypertension [23]. The amount of urate in the blood depends on various lifestyle factors but is primarily determined by genetic factors that regulate the level of endogenous urate biosynthesis and the rate of uric acid excretion. Although the underlying molecular mechanisms are not yet fully understood, it seems that a reduction in endothelial nitric oxide production and stimulation of the vascular renin-angiotensin system, which increases angiotensin II production and subsequently vascular smooth muscle cell proliferation and oxidative stress, may mediate the hypertensive effects of uric acid [24]. The present data confirm the alteration in the purine metabolism during hypertension, and recommend preventive hypertension monitoring strategies in people with increased uric acid levels.

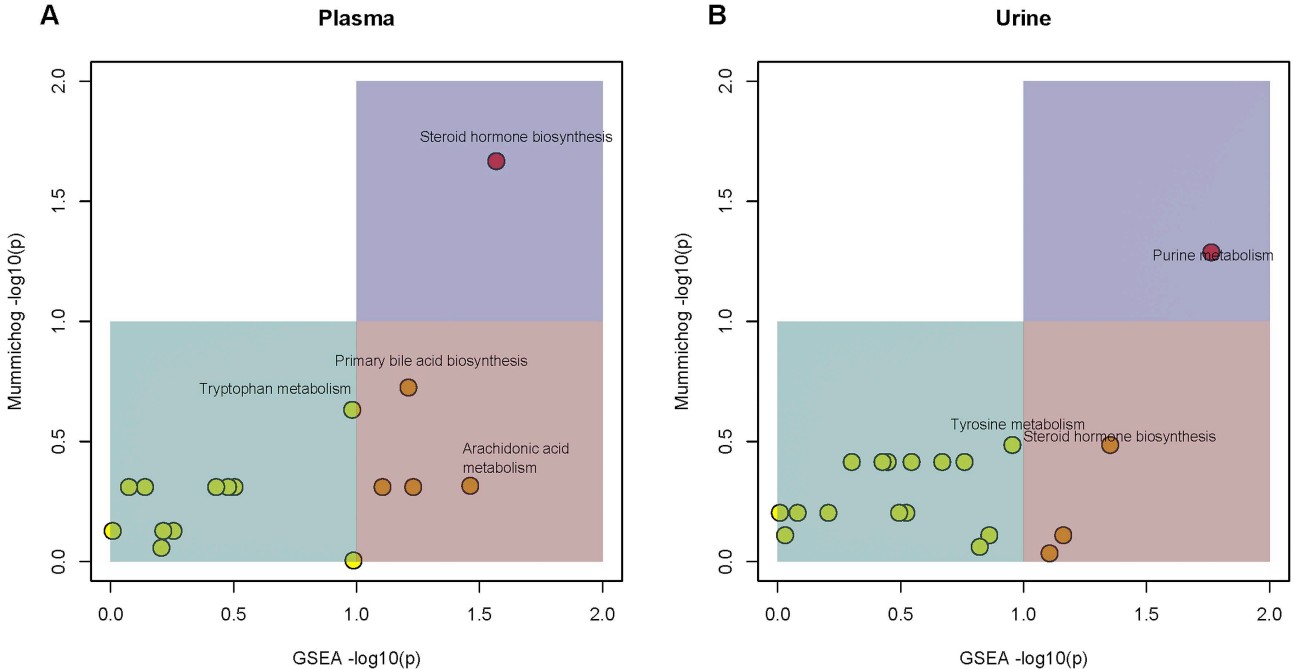

**Fig 3.** *Mummichog* pathway enrichment analysis of plasma (A) and urine (B) metabolites.

In the present study, an elevation of citrate was detected in urine samples but not in plasma, suggesting that increased excretion may be a consequence of altered renal metabolism instead rather than systemic metabolism. Aconitase catalyzes the conversion of citrate to isocitrate and is inhibited by uric acid; therefore, the elevation of uric acid observed in SHR might be responsible for, or at least contribute to, the accumulation of citric acid in renal cells, which flows out of the mitochondria to the cytosol and is then excreted in the urine. In contrast, other studies have reported lower excretion of citric acid in urine from 11–20 weeks old SHR compared with age-matched WKY rats [25–27]. Therefore, a conclusion regarding citric acid metabolism in SHR cannot be drawn, and further investigation are needed to determine its relationship with hypertension.

Higher amounts of urine pseudouridine were found in the SHRs compared to WKY. Pseudouridine is the most widely distributed post-transcriptionally modified nucleotide formed by pseudouridine synthases in RNA which involved in epigenetic regulation of gene expression [28], including mitochondrial gene expression [29]. The knowledge of the role of mitochondria in the regulation of the smooth muscle phenotype and differentiation is increasing [30], and pseudouridine has been proposed as a potential biomarker of cardiac dysfunction [29,31,32]. However, to the best of our knowledge, no evidence has connected the epigenetic mechanism of pseudouridine with the pathogenesis of hypertension, and future studies are needed in this field.

Accumulating evidence suggests a key role for the gut microbiota in essential and experimental hypertension in animals and human [33–36]. Hypertensive animals and patients have decreased microbial richness, diversity, and evenness, and an increased *Firmicutes/Bacteroidetes* ratio [37]. It has been demonstrated that hypertension contributes to unhealthy shifts of the gut microbiota [38]. The protective effects of different antihypertensive peptides have been attributed, at least in part, to their ability to modulate gut microbiota dysbiosis [39,40]. In SHR, differences in gut microbial composition have been reported compared with WKY rats, and blood pressure was regulated in SHR following cross fecal transplantation from WKY, while normotensive animals became hypertensive when receiving fecal transplantation from SHR animals [41].

Similarly, our results show differences in dietary-derived gut microbial metabolites, namely hippuric acid and diverse sulfated and glucuronidated products, between hypertensive and normotensive rats, which are not due to differences in the type and amount of diet since all animals ate the same.

Cholic acid is a discriminant metabolite that is lower in the plasma of SHR than that in WKY rats. In addition, *mummichog* algorithms have recognized the biosynthesis of primary bile acids as a metabolic pathway affected by SHR, including four taurine-derived salts. Other authors have suggested a protective effect of taurine-conjugated bile acids against the development of hypertension and have reported that bile acid conjugation was inversely associated with SBP. They described a distinct clustering of taurine-conjugated bile acids that was less abundant in genetically hypertensive Dahl rats than in normotensive animals, independent of salt consumption. Furthermore, the accumulation of microbiota-derived taurine-conjugated bile acids was associated with lower blood pressure, as well as with specific microbiota taxa [42]. Regarding tyrosine metabolism, p-cresol conjugates were lower in hypertensive SHR than in WKY rats, specifically plasma levels of p-cresol sulfate and glucuronide and urinary p-cresol glucuronide. In addition, a hypotensive effect of taurine in SHR, associated with an increase in p-cresol derivatives, has been observed, and may be mediated through the modulation of intestinal microbiota metabolism [25]. We also observed reduced levels of urinary hippuric acid, another gut microbiota-derived metabolite [43], in agreement with data described by Akira et al.[44] using an NMR-based metabonomic approach. Although one limitation of the present study is that gut microbiota composition was not analyzed, our findings suggest that gut microbiota dysbiosis is associated with hypertension, and further identify that derived primary liver bile acid metabolism is affected during hypertension. However, further studies are required to elucidate the underlying mechanisms.

Although other authors have reported changes in the metabolic profiles associated with hypertension in animal models and humans [31], the relationship with glycerophospholipids remains controversial and appears to depends on the specific lipid class [6]. In 2022, Liu et al. identified eight blood pressure-related plasma phospholipids (six phosphatidylethanolamines and two phosphatidylcholines) with predictive value for hypertension risk [45], whereas other diacyl-phosphatidylcholines (C38:4 and C38:3) were associated with enhanced hypertension complications [46]. Onuh and Allani [6] reported that higher levels of different acyl-alkyl-phosphatidylcholines (C42:4 and C44:3) were associated with lower fatal hypertension in humans, possibly because of the protective antioxidant and inflammatory activities of these metabolites. Here, we identified lower plasma levels of some lysoglycerophospholipids in hypertensive rats than in normotensive rats. Decreased levels of lysophospatidylcholines have been observed in several inflammatory-based diseases, including pulmonary arterial hypertension, and are associated with increased mortality risk [47]. However, Jiang et al.[48] proposed lysophospatidylcholines as a biomarker of hypertension, suggesting that LDL oxidation may promote their generation, thereby interfering with nitric oxide production and contributing to hypertension. Therefore, further investigations in humans with hypertension are required to confirm certain assumptions related to glycerophospholipid metabolism.

## Conclusion

The investigation of metabolites associated with hypertension and their consequences is of great interest for elucidating the underlying molecular mechanisms and assessing the risk of pathological consequences. Our research suggests an association between gut microbiota-derived metabolites, bile acids, steroid hormones, and purine metabolism with hypertension, which may represent either risk factors or consequences of the disease. Therefore, more investigations are required to clarify their roles.

## Supporting information

**S1 File. Scores plot and permutation test plots from OPLS-DA models in urine and plasma.** Spectral comparison between measured MS/MS of the most discriminant metabolites in samples and match in libraries.
(DOCX)

## Author contributions

**Conceptualization:** Celia Rodríguez-Pérez, Estefanía Sánchez-Rodríguez, Ángel Gil, María D. Mesa.

**Data curation:** Celia Rodríguez-Pérez, Estefanía Sánchez-Rodríguez.

**Formal analysis:** Oscar Daniel Rangel Huerta, Celia Rodríguez-Pérez, Alejandra Vázquez-Aguilar, Estefanía Sánchez-Rodríguez.

**Funding acquisition:** Celia Rodríguez-Pérez, María D. Mesa.

**Investigation:** Celia Rodríguez-Pérez, Estefanía Sánchez-Rodríguez, Caridad Díaz, Félix Vargas, María D. Mesa.

**Methodology:** Oscar Daniel Rangel Huerta, Celia Rodríguez-Pérez, Alejandra Vázquez-Aguilar, Estefanía Sánchez-Rodríguez, Caridad Díaz, Félix Vargas, María D. Mesa.

**Project administration:** Celia Rodríguez-Pérez, María D. Mesa.

**Resources:** Celia Rodríguez-Pérez.

**Software:** Oscar Daniel Rangel Huerta.

**Supervision:** Celia Rodríguez-Pérez, Ángel Gil, María D. Mesa.

**Validation:** Oscar Daniel Rangel Huerta.

**Visualization:** Oscar Daniel Rangel Huerta.

**Writing – original draft:** Celia Rodríguez-Pérez, Alejandra Vázquez-Aguilar.

**Writing – review & editing:** Oscar Daniel Rangel Huerta, Estefanía Sánchez-Rodríguez, Ángel Gil, Caridad Díaz, María D. Mesa.

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
