## [Decision Letter · Decision Letter 0]

15 Dec 2025

Dear Dr. Rangel Huerta,

We look forward to receiving your revised manuscript.

Kind regards,

Rami Salim Najjar, Ph.D.

Academic Editor

PLOS One

Journal Requirements:

2. To comply with PLOS One submissions requirements, in your Methods section, please provide additional information regarding the experiments involving animals and ensure you have included details on (1) methods of sacrifice, (2) methods of anesthesia and/or analgesia, and (3) efforts to alleviate suffering.

4. Please note that funding information should not appear in any section or other areas of your manuscript. We will only publish funding information present in the Funding Statement section of the online submission form. Please remove any funding-related text from the manuscript.

Reviewer's Responses to Questions

**Comments to the Author**

1. Is the manuscript technically sound, and do the data support the conclusions?

Reviewer #1: No

Reviewer #2: No

Reviewer #3: Partly

2. Has the statistical analysis been performed appropriately and rigorously?

Reviewer #1: No

Reviewer #2: No

Reviewer #3: Yes

3. Have the authors made all data underlying the findings in their manuscript fully available?

Reviewer #1: No

Reviewer #2: Yes

Reviewer #3: Yes

4. Is the manuscript presented in an intelligible fashion and written in standard English?

Reviewer #1: No

Reviewer #2: Yes

Reviewer #3: Yes

Reviewer #1: The manuscript is inadequately prepared, with serious inconsistencies between the cited figures in the main text, their captions, and the corresponding content of figures. Notably, the figures presented do not align with or support the results. Even after the authors reload the figures, the study demonstrates limited novelty and insufficient scholarly contribution. Furthermore, the experimental results show minimal divergence from those reported in previous research. Given these concerns, publication of this work in the journal is not recommended.

Reviewer #2: This study focuses on non-targeted metabolomic differences in plasma and urine between 16-week-old spontaneously hypertensive rats (SHR) and age-matched normotensive Wistar-Kyoto (WKY) rats, representing a research direction of some value. However, the manuscript exhibits several critical flaws that preclude its publication.

1.Figures 1, 2, and 3 are completely irrelevant to the article, and crucial representative figures are missing.

2.Low Confidence in Differential Metabolite Identification: Only 16 plasma and 13 urinary differential metabolites were annotated. This represents an extremely low proportion of the total differential features screened (59 in plasma and 111 in urine). The failure to identify a substantial number of potential key metabolites undermines the completeness of the study.

3.Superficial and Disconnected Results Analysis: The interpretation of the biological significance of the differential metabolites remains superficial. There is a lack of in-depth analysis linking the findings to the pathophysiological mechanisms of hypertension. For instance, while differences in gut microbiota-derived metabolites are mentioned, the discussion fails to explore the specific mechanisms linking these metabolites to blood pressure regulation.

4.The Discussion largely merely repeats the Results section and does not adequately contrast the findings of this study with previous research, discussing both consistencies and discrepancies with the existing literature.

Reviewer #3: Reviewer’s Comments

Journal: PLOS One

Article: New insights into the plasma and urinary metabolomic signatures of spontaneously

hypertensive rats

Authors: Huerta et al.

Overview

This work aimed to explore the plasma and urinary non-targeted metabolic profile of 16-week-old spontaneously hypertensive rats (SHR) to identify new metabolomic profiles associated with hypertensive phenotypical characteristics. This study is very important to create better understanding of the pathogenesis of diseases like hypertension and other cardiovascular diseases and their complications. Additionally, it will also identify biomarkers for these conditions as well as associated metabolic mechanisms and pathways by which they modulate these conditions. The study therefore, hoped to offer alternative approaches for treatment of hypertension and cardiovascular diseases. The concept was well thought out and the various analyses were clearly designed to address the objectives of the study. The results obtained and reasons adduced for the reported effects and conclusions drawn are succinct to support the mechanisms. However, there were so many flaws in the study in its current form that should be resolved to make it publishable.

Major comments

1. The authors did not state the age at which the animals were purchased? Were they purchased at 5 weeks of age and kept until 16 weeks of age before termination and sample collection? Or, were they purchased at 16 weeks which was the time of termination and sample collection?

2. How long was the “SBP was monitored to evaluate the state of the disease?”

3. Was SBP measurement done in conscious or unconscious animals? If in conscious state, what measures were taken to restrain the animals so that BP readings are not affected by their unrestrained movements? If in unconscious state, what was the procedure for rendering the animals unconscious? The method was not detailed enough to give the reader a clearer picture for replication.

4. How ideal is the Mummichog software for biomarker identification since it only identifies significantly enriched metabolic pathways to create hypotheses by bypassing the need to fully identify every single metabolite? Without fully identifying individual metabolites in the system, it will be difficult to have a clear picture of metabolites that may be significantly changing and could be considered biomarkers for the disease or condition.

5. For the plasma and urine metabolomics samples preparation, were internal standards used, and if so, what was the internal standard and at what concentration/sample?

6. In the results section (3.1), the authors discussed the heart weight, left ventricular weight, water intake, diuresis, and heart rates and other organ weights. These were not discussed in the methodology section previously to give an idea of what was done in the various methods, especially with regards to organ collection.

7. Page 11, last paragraph, “The supervised multivariate analysis OPLS-DA indicated a strong separation between the two groups (data not shown).” Why would the data for such an important information not be shown to give readers an idea of the separation between the 2 groups?

8. Page 24, Table 1; why the middle line between heart rate and tibia length?

9. Page 25, Table 2; the authors need to create a column to shown which of the metabolites were either upregulated or down-regulated.

10. Page 26, Table 3; the authors need to create a column to shown which of the metabolites were either upregulated or down-regulated.

11. I am not sure if figure 1 was for this particular study, as it resembles a clinical study more than the animal study.

12. The figures are not properly aligned with the titles. Check that they represent what they claimed to be.

13. Notable pathways and mechanisms for hypertension especially oxidative stress and inflammation were not reported by this study, suggesting a limitation in the Mummichog software used and therefore, the need for more high throughput software to validate the outcomes.

Minor comments

1. There are some typographical errors throughout the texts that needs revisions. Since they manuscript lacked line numbers, it was difficult outlining specific errors.

**Do you want your identity to be public for this peer review?** For information about this choice, including consent withdrawal, please see our For information about this choice, including consent withdrawal, please see our Privacy Policy .

Reviewer #1: No

Reviewer #2: No

Reviewer #3: **Yes:** John OnuhJohn Onuh

---

## [Author Response · Author response to Decision Letter 1]

22 Jan 2026

Reviewer’s Comments

Journal: PLOS One

Article: New insights into the plasma and urinary metabolomic signatures of spontaneously

hypertensive rats

Authors: Huerta et al.

Overview

This work aimed to explore the plasma and urinary non-targeted metabolic profile of 16-week-old spontaneously hypertensive rats (SHR) to identify new metabolomic profiles associated with hypertensive phenotypical characteristics. This study is very important to create better understanding of the pathogenesis of diseases like hypertension and other cardiovascular diseases and their complications. Additionally, it will also identify biomarkers for these conditions as well as associated metabolic mechanisms and pathways by which they modulate these conditions. The study therefore, hoped to offer alternative approaches for treatment of hypertension and cardiovascular diseases. The concept was well thought out and the various analyses were clearly designed to address the objectives of the study. The results obtained and reasons adduced for the reported effects and conclusions drawn are succinct to support the mechanisms. However, there were so many flaws in the study in its current form that should be resolved to make it publishable.

Major comments

1. The authors did not state the age at which the animals were purchased? Were they purchased at 5 weeks of age and kept until 16 weeks of age before termination and sample collection? Or, were they purchased at 16 weeks which was the time of termination and sample collection?

R: Thank you for your comments. Rats were purchased at 8 weeks of age and kept until 16 weeks of age before termination and sample collection. We have clarified this issue in Page 6, line 64.

2. How long was the “SBP was monitored to evaluate the state of the disease?”

R: Thank you for your comments, SBP was monitored to check the evolution of the disease. We have clarified this issue in Page 7, line 78-84.

3. Was SBP measurement done in conscious or unconscious animals? If in conscious state, what measures were taken to restrain the animals so that BP readings are not affected by their unrestrained movements? If in unconscious state, what was the procedure for rendering the animals unconscious? The method was not detailed enough to give the reader a clearer picture for replication.

R: Systolic blood pressure (SBP), diastolic blood pressure (DBP), and heart rate (HR) are measured in conscious rats using a non-invasive tail-cuff plethysmography system (LE 5001 Pressure Meter, Letica SA, Barcelona, Spain). Animals are first habituated to the procedure on several days prior to the experiment to minimize stress-related artifacts. On the day of measurement, rats are placed in individual restrainers that allow immobilization without the use of anesthesia. The tail is gently extended through the rear opening of the restrainer. This information has been included in Page 7, line 78-84.

4. How ideal is the Mummichog software for biomarker identification since it only identifies significantly enriched metabolic pathways to create hypotheses by bypassing the need to fully identify every single metabolite? Without fully identifying individual metabolites in the system, it will be difficult to have a clear picture of metabolites that may be significantly changing and could be considered biomarkers for the disease or condition.

R: The results of mummichog are very relevant to provide an insight into the pathways that might be altered. We used this approach for double crossing with the univariate/multivariate analysis. As we had some potential annotations but we were not able to run MS/MS experiments to confirm the identity, we used the combined information to increase the confidence in our annotations.

5. For the plasma and urine metabolomics samples preparation, were internal standards used, and if so, what was the internal standard and at what concentration/sample?

R: This information was included in page 8 as follows: “Five microliters of internal standard (IS) consisting of a mixture of roxithromycin 3 ppm, tryptophan-D5 5 ppm, hydroxydiclofenac 5 ppm, warfarin 3 ppm, ibuprofen 5 ppm, and bisphenol 3 ppm , were added prior the extraction”.

6. In the results section (3.1), the authors discussed the heart weight, left ventricular weight, water intake, diuresis, and heart rates and other organ weights. These were not discussed in the methodology section previously to give an idea of what was done in the various methods, especially with regards to organ collection.

R: Methodology details has been included:

“Food and water intake were monitored daily by weighing the remaining food or by measuring the remaining volume of water, respectively, allowing the calculation of daily food and water consumption.” (in page 7, lines 85-88).

“In addition, after euthanasia, necropsy was performed and the heart, kidneys, liver, and tibia were collected and their absolute weights recorded to evaluate animal development and organ-specific involvement.” (in page 7, lines 90-93).

7. Page 11, last paragraph, “The supervised multivariate analysis OPLS-DA indicated a strong separation between the two groups (data not shown).” Why would the data for such an important information not be shown to give readers an idea of the separation between the 2 groups?

R: In this context, the plots are omitted, but the model's quality metrics are included. We chose not to present the scores plot because we consider the metrics to be more meaningful; these models often overfit, even with random data, making the plots less informative than the metrics themselves. The phrase has been rewritten (L198)

8. Page 24, Table 1; why the middle line between heart rate and tibia length?

R: Thank you for the comment. This was an error that has been eliminated.

9. Page 25, Table 2; the authors need to create a column to shown which of the metabolites were either upregulated or down-regulated.

R: The column is now included to facilitate the interpretation of the results.

10. Page 26, Table 3; the authors need to create a column to shown which of the metabolites were either upregulated or down-regulated.

R: The column is now included to facilitate the interpretation of the results.

11. I am not sure if figure 1 was for this particular study, as it resembles a clinical study more than the animal study.

R: The correct figure is now attached.

12. The figures are not properly aligned with the titles. Check that they represent what they claimed to be.

R: The correct figure is now attached.

13. Notable pathways and mechanisms for hypertension especially oxidative stress and inflammation were not reported by this study, suggesting a limitation in the Mummichog software used and therefore, the need for more high throughput software to validate the outcomes.

R: The use of Mummichog in our study is based on its ability to infer metabolic pathway enrichment directly from global metabolomics data without requiring prior identification of all metabolites, which is particularly suitable for exploratory analyses based on high-resolution LC-MS. The absence of pathways classically associated with hypertension, such as oxidative stress or inflammation, does not necessarily imply that these biological processes are not involved, but may reflect inherent limitations in the coverage of metabolites in reference databases, the representation of these pathways, and the statistical criteria for enrichment. In this context, pathways with a small number of annotated metabolites or insufficiently represented m/z signals may not reach statistical significance, even when the underlying processes are biologically active. Therefore, this limitation is not unique to Mummichog, but common to enrichment-based pathway analysis approaches in untargeted metabolomics studies. We agree with the reviewer that the application of complementary tools and more in-depth approaches, including methods based on identified metabolites and cross-validation with alternative software, could strengthen the biological interpretation and will be considered in future studies.

Minor comments

1. There are some typographical errors throughout the texts that needs revisions. Since they manuscript lacked line numbers, it was difficult outlining specific errors.

R: The manuscript has been carefully revised, and several corrections were made across it.

---

## [Decision Letter · Decision Letter 1]

5 Feb 2026

Dear Dr. Rangel Huerta,

Thank you for submitting your manuscript to PLOS ONE. After careful consideration, we feel that it has merit but does not fully meet PLOS ONE’s publication criteria as it currently stands. Therefore, we invite you to submit a revised version of the manuscript that addresses the points raised during the review process.

We look forward to receiving your revised manuscript.

Kind regards,

Rami Salim Najjar, Ph.D.

Academic Editor

PLOS One

Journal Requirements:

Reviewers' comments:

Reviewer's Responses to Questions

**Comments to the Author**

Reviewer #2: (No Response)

Reviewer #3: All comments have been addressed

2. Is the manuscript technically sound, and do the data support the conclusions?

Reviewer #2: No

Reviewer #3: Yes

3. Has the statistical analysis been performed appropriately and rigorously?

Reviewer #2: No

Reviewer #3: Yes

4. Have the authors made all data underlying the findings in their manuscript fully available?

Reviewer #2: No

Reviewer #3: Yes

5. Is the manuscript presented in an intelligible fashion and written in standard English?

Reviewer #2: Yes

Reviewer #3: Yes

Reviewer #2: Major Concerns

Unjustified Internal Standard (IS) Selection (Page 8, line 101; Response to Comment 5):

The IS mixture used is atypical for a non-targeted metabolomics study of endogenous metabolites. Its composition (mainly pharmaceuticals with only one stable isotope-labeled standard) has poor physicochemical resemblance to the key metabolite classes identified (e.g., bile acids, glycerophospholipids). The authors must provide a detailed justification in the Methods or Discussion for this specific choice, explaining how each component monitors technical variance relevant to their analyte panel. Furthermore, they should critically evaluate and cite the age and context of any literature that employs a similar IS protocol, acknowledging this as a potential limitation for quantitative rigor if modern best practices were not followed.

Insufficient Metabolite Identification Confidence:

There is a contradiction between the response (stating MS/MS was not run for confirmation) and the data presented in Tables 2 & 3 (listing "Fragments"). The confidence level for all metabolite annotations must be explicitly stated (e.g., following Metabolomics Standards Initiative guidelines). Supporting MS/MS spectra for key annotated metabolites should be provided in the supplementary information.

Deficient Data Presentation and Visualization:

OPLS-DA Plots: The decision to omit score plots is unacceptable. Visual assessment of group separation is essential. The OPLS-DA score plots and corresponding permutation test plots for both plasma and urine models must be provided (Main or Supplementary Figures).

S-Plots (Figure 2): The most discriminating metabolites (variables) should be clearly labeled in the S-plots to align with the data in Tables 2 & 3, enhancing reader interpretation.

PCA Plot (Figure 1A): The presented plasma PCA plot shows incomplete separation between groups. This should be acknowledged and discussed in the context of the model's performance and the stronger separation suggested by the OPLS-DA parameters.

Tables 2 & 3: These tables appear truncated or misformatted in the manuscript file. They must be checked and presented completely and clearly.

Specific Editorial and Technical Corrections

Reference Callouts: Several references are listed in the text (e.g., 1, 2, 3) but are not properly cited with superscript numbers in the relevant sentences. All in-text citations must be correctly placed.

Formatting Consistency: The manuscript exhibits inconsistent paragraph indentation. Formatting must be unified according to the journal's style guide.

Text Errors: Please remove the extra space in line 102.

Reviewer #3: The authors have addressed all my previous concerns and the manuscript is now in a better shape to be accepted for publication. I therefore, have no concern anymore regarding the quality of the manuscript in its present form.

**Do you want your identity to be public for this peer review?** For information about this choice, including consent withdrawal, please see our For information about this choice, including consent withdrawal, please see our Privacy Policy .

Reviewer #2: No

Reviewer #3: **Yes:** Dr. John O. OnuhDr. John O. Onuh

---

## [Author Response · Author response to Decision Letter 2]

13 Feb 2026

Reviewer #2: Major Concerns

Unjustified Internal Standard (IS) Selection (Page 8, line 101; Response to Comment 5):

The IS mixture used is atypical for a non-targeted metabolomics study of endogenous metabolites. Its composition (mainly pharmaceuticals with only one stable isotope-labeled standard) has poor physicochemical resemblance to the key metabolite classes identified (e.g., bile acids, glycerophospholipids). The authors must provide a detailed justification in the Methods or Discussion for this specific choice, explaining how each component monitors technical variance relevant to their analyte panel. Furthermore, they should critically evaluate and cite the age and context of any literature that employs a similar IS protocol, acknowledging this as a potential limitation for quantitative rigor if modern best practices were not followed.

R: Thank you for the observation. We included internal standards (IS) to monitor instrument stability and to normalize sample-to-sample variation, not for quantitative purposes. To avoid interference with sample signals, we specifically selected IS compounds whose chromatographic peaks do not overlap with signals observed in our samples. This rationale has been validated in our prior studies, where the chosen IS performed as intended without overlapping sample features and without compromising data integrity. Reference 9 has been included in page 8.

Insufficient Metabolite Identification Confidence:

There is a contradiction between the response (stating MS/MS was not run for confirmation) and the data presented in Tables 2 & 3 (listing "Fragments"). The confidence level for all metabolite annotations must be explicitly stated (e.g., following Metabolomics Standards Initiative guidelines). Supporting MS/MS spectra for key annotated metabolites should be provided in the supplementary information.

R: Unfortunately, the response was not correctly formulated. We did not run MS/MS of chemical standards to validate the MS/MS from our analysis. Spectral Data is now attached as Supplemental material and level of confidence is included in the tables.

Deficient Data Presentation and Visualization:

OPLS-DA Plots: The decision to omit score plots is unacceptable. Visual assessment of group separation is essential. The OPLS-DA score plots and corresponding permutation test plots for both plasma and urine models must be provided (Main or Supplementary Figures).

In this context, the plots were omitted, but the model's quality metrics are included. We chose not to present the scores plot because we consider the metrics to be more meaningful; these models often overfit, even with random data, making the plots less informative than the metrics themselves. However, as suggested by the reviewer, the phrase has been rewritten (L198) and the plots are included as supplemental material.

S-Plots (Figure 2): The most discriminating metabolites (variables) should be clearly labeled in the S-plots to align with the data in Tables 2 & 3, enhancing reader interpretation.

R: The information has been attached.

PCA Plot (Figure 1A): The presented plasma PCA plot shows incomplete separation between groups. This should be acknowledged and discussed in the context of the model's performance and the stronger separation suggested by the OPLS-DA parameters.

R: PCA is a non-supervised analysis where no label is assigned to the samples, therefore is a valid approach to explore if clustering is present through an unbiased technique. The separation between groups is clear and complete although there is variation within the groups. The OPLS-DA analysis as a supervised approach is characterized by isolation of any orthogonal variation. In this case, this within group variation is properly handed by the modeling and therefore the group variation is maximized.

Tables 2 & 3: These tables appear truncated or misformatted in the manuscript file. They must be checked and presented completely and clearly.

R: Tables are now adjusted to a proper format

Specific Editorial and Technical Corrections

Reference Callouts: Several references are listed in the text (e.g., 1, 2, 3) but are not properly cited with superscript numbers in the relevant sentences. All in-text citations must be correctly placed.

R: The references have been checked and modified accordingly.

Formatting Consistency: The manuscript exhibits inconsistent paragraph indentation. Formatting must be unified according to the journal's style guide. Text Errors: Please remove the extra space in line 102.

R: The formatting has been checked and changed accordingly.

---

## [Decision Letter · Decision Letter 2]

25 Feb 2026

New insights into the plasma and urinary metabolomic signatures of spontaneously hypertensive rats

PONE-D-25-40750R2

Dear Dr. Rangel Huerta,

We’re pleased to inform you that your manuscript has been judged scientifically suitable for publication and will be formally accepted for publication once it meets all outstanding technical requirements.

Kind regards,

Rami Salim Najjar, Ph.D.

Academic Editor

PLOS One

Additional Editor Comments (optional):

Reviewers' comments:

Reviewer's Responses to Questions

**Comments to the Author**

Reviewer #2: (No Response)

2. Is the manuscript technically sound, and do the data support the conclusions?

Reviewer #2: (No Response)

3. Has the statistical analysis been performed appropriately and rigorously?

Reviewer #2: (No Response)

4. Have the authors made all data underlying the findings in their manuscript fully available?

Reviewer #2: (No Response)

5. Is the manuscript presented in an intelligible fashion and written in standard English?

Reviewer #2: (No Response)

Reviewer #2: (No Response)

**Do you want your identity to be public for this peer review?** For information about this choice, including consent withdrawal, please see our For information about this choice, including consent withdrawal, please see our Privacy Policy .

Reviewer #2: No

---

## [Editor Report · Acceptance letter]

PONE-D-25-40750R2

PLOS One

Dear Dr. Rangel Huerta,

I'm pleased to inform you that your manuscript has been deemed suitable for publication in PLOS One. Congratulations! Your manuscript is now being handed over to our production team.

Kind regards,

on behalf of

Dr. Rami Salim Najjar

Academic Editor

PLOS One